# Effect of Micro-/Nanoparticle Hybrid Hydrogel Platform on the Treatment of Articular Cartilage-Related Diseases

**DOI:** 10.3390/gels7040155

**Published:** 2021-09-27

**Authors:** Xu Han, Yongzhi Wu, Yue Shan, Xu Zhang, Jinfeng Liao

**Affiliations:** National Clinical Research Centre for Oral Diseases, State Key Laboratory of Oral Diseases, West China Hospital of Stomatology, Sichuan University, Chengdu 610041, China; 1293145195@qq.com (X.H.); wuyongzhiscu@163.com (Y.W.); 294113271@qq.com (Y.S.); zx2020224030006@163.com (X.Z.)

**Keywords:** microparticles, nanoparticles, hydrogel, articular cartilage related diseases

## Abstract

Joint diseases that mainly lead to articular cartilage injury with prolonged severe pain as well as dysfunction have remained unexplained for many years. One of the main reasons is that damaged articular cartilage is unable to repair and regenerate by itself. Furthermore, current therapy, including drug therapy and operative treatment, cannot solve the problem. Fortunately, the micro-/nanoparticle hybrid hydrogel platform provides a new strategy for the treatment of articular cartilage-related diseases, owing to its outstanding biocompatibility, high loading capability, and controlled release effect. The hybrid platform is effective for controlling symptoms of pain, inflammation and dysfunction, and cartilage repair and regeneration. In this review, we attempt to summarize recent studies on the latest development of micro-/nanoparticle hybrid hydrogel for the treatment of articular cartilage-related diseases. Furthermore, some prospects are proposed, aiming to improve the properties of the micro-/nanoparticle hybrid hydrogel platform so as to offer useful new ideas for the effective and accurate treatment of articular cartilage-related diseases.

## 1. Introduction

Cartilage, including hyaline cartilage, fibrous cartilage, and elastic cartilage, is distributed throughout the human body and plays various roles, including lubricating joints, bearing loads, and supporting structures [1]. Nevertheless, plenty of diseases occur in cartilage that are seriously harmful to humans. Due to the unavoidable constant motion of the human body, articular cartilage in the joints is more easily damaged. Due to the aging of the population and modern environmental factors, this problem is becoming increasingly serious [2]. Osteoarthritis (OA), a degenerative disease that causes progressive degeneration of the articular cartilage [3], wears on the body’s structures, afflicting many patients with chronic pain [4] for many years [5]. In addition, rheumatoid arthritis (RA) [6] also results in severe cartilage damage, which affects multiple joints in the body. Moreover, inflammatory arthropathies [7], cartilaginous tumors [8], chondrocalcinosis [9], developmental disorders [10], relapsing polychondritis (RP) [11], and osteochondritis dissecans [12] cause cartilage injury, dysfunction, and severe pain. These diseases commonly involve joints such as the knees, ankles, knuckles, and shoulder joints, as well as temporomandibular joints. It is ineluctable to have trouble with walking, finger, and shoulder movement, and chewing food to different degrees as a daily activity requires the participation of these joints. Figure 1 shows the internal structure of articular cartilage defects in the temporomandibular joint (TMJ) and knee joint.

Due to the features of negatively charged and lacking blood vessels, nerves, and the lymphatic system, it is very difficult for cartilage-related diseases to be cured. At present, drug therapy [13,14] and surgical treatment [15,16] are the main methods to treat articular cartilage-related diseases. However, these clinical treatments are circumscribed and show limited effects. The majority of drugs are used to relieve symptoms such as severe pain [15] and inflammation rather than for an etiological cure. In addition, the function of current drugs in the clinical setting has been proven to be unsatisfactory. Although intra-articular injection is proposed to have a positive effect, unsolved problems still exist, such as safety [17], limited drug effects [18], and poor penetration effects [19]. It is likely to accelerate the progression of joint diseases and damage to joints [17]. Furthermore, surgical treatment tends to cause large wounds and infection, resulting in a limited age for surgery in the clinic [20]. Obviously, tissue regeneration is fairly difficult to entirely realize by the rare methods mentioned above [21]. The lack of an appropriate drug delivery system has already puzzled scientists for a long time. Systemic administration may lead to severe side effects and has a weak effect on the pathological area. However, local injection requires several administrations to ensure a long-lasting effect, and injectable drugs are limited [22]. 

With the rapid development of materials science, molecular biology, and biomedicine, new choices to improve the aforementioned problem have arisen. Specifically, microparticles [23], nanoparticles [24], and hydrogel [25] are suitable for the tissue regeneration and treatment of localized cartilage diseases due to their capability for controlled drug release, enhanced lubrication, and biocompatibility [26,27,28,29]. Hydrogel offers an appropriate growth environment for chondrocytes and has multiple performances, such as injectability [30], biocompatibility [31], and capability of being environment-sensitive [32] and self-healing [33,34,35]. At the same time, we cannot ignore that they still have limited functions and disadvantages. For example, poor mechanical properties of injectable hydrogel affect its use in joints. The short release time of growth factors in micro-/nanoparticles influences their effect on cartilage repair. Therefore, it is better to combine them into a micro-/nanoparticle hybrid hydrogel platform in order to take advantage of their strengths and compensate for their shortcomings. A great number of researchers have already focused on this for research into multi-field as well as cartilage-related diseases. Some studies show that nanospheres can enhance lubrication [36], ductility, and self-recovery of hydrogel [37]. Microparticles are found to make up for the deficiency of hydrogel in mechanical and stable properties. Meanwhile, the capability of controlled drug release is improved. In addition, the loading efficacy and drug types can be increased [38,39,40]. Current studies have proved the superiority and feasibility of micro-/nanoparticle hybrid hydrogel for the therapy of articular cartilage-related diseases. 

Herein, we focus on the treatment of articular cartilage-related diseases based on the micro-/nanoparticle hybrid hydrogel platform in recent years. The effects of the platform are discussed in two parts, including symptomatic treatment and the treatment of articular cartilage defects. Finally, we analyze challenges that the micro-/nanoparticle hybrid hydrogel platform faces in the treatment of articular cartilage-related diseases and proposed prospects in research and clinical application in the future. 

## 2. The Preparation and Main Properties of the Micro-/Nanoparticle Hybrid Hydrogel Platform

There is a variety of micro-/nanoparticles made from multiple materials, which have diverse features and play different roles. Alginate [41], chitosan [27,42], gelatin [43], metal [44], and silk fibroin [45] are representative materials to prepare microparticles or nanoparticles in current research [46,47,48,49]. Different methods are used to prepare porous [50], hollow [51,52], and fuzzy [53] micro-/nanoparticles. Hydrogel, made from chitosan [54], alginate [54], gelatin [55], hyaluronic acid (HA) [56], and cellulose [57], as well as synthesized polymers such as polyethylene glycol (PEG) [58], poly vinyl alcohol (PVA) [33,59], gelatin methacryloyl (GelMA) [60], and poly caprolactone (PCL) [61], have diverse properties. Additionally, bi-layered hydrogel was also designed and showed potential in the repair of osteochondral defects [62,63]. Their abilities to be degradable [64]; self-healing [65]; self-adapting [66]; injectable [57,67]; tunable mechanical [68] and stimuli-responsive, such as pH-responsive [57,69]; thermal-responsive [70]; light-responsive [71]; and magnetic-responsive [72] are crucial for hydrogel research and are expected to be applied in clinical practice. Figure 2 shows the variety of micro-/nanoparticles and hydrogels according to their diverse properties.

Aiming to obtain hydrogels and micro-/nanoparticles with various properties made from different materials requires preparation using several methods. As for hydrogel preparation, physical crosslinking and chemical crosslinking are the main methods, including the freeze–heating cycle [73], Schiff-based reaction [74], and Michael addition [75]. To prepare microparticles, methods such as propylene carbonate emulsification–extraction [76], the membrane emulsification technique [77], liquid jet breakup [78], the electrostatic microdroplet method [79], spray drying, and chemical crosslinking [80] are used by researchers. Furthermore, some techniques are applied to prepare nanoparticles, including emulsification [81], nanoprecipitation [82], the double emulsion solvent evaporation method [83], and the microfluidic technique [84].

In order to understand the benefits of both micro-/nanoparticles and hydrogel, the micro-/nanoparticle hybrid hydrogel platform is being widely studied at present. First, microparticles and nanoparticles were prepared and stowed. Then, to combine micro-/nanoparticles with hydrogel, microparticles and nanoparticles were completely dispersed in a solution to make hydrogel. Finally, we discovered that micro-/nanoparticles were embedded in the hydrogel [38,85,86,87,88,89].

The micro-/nanoparticle hybrid hydrogel platform possesses varying properties due to different selections of materials and the characteristics of specific research interests. Its advantages are mainly as follows: First, the controlled and sustained release property is significant for drug delivery [90,91]. It is beneficial to avoid frequent multiple doses and damage caused by large doses. Second, multiple loads can be transported together by loading in micro-/nanospheres and hydrogel [92]. Figure 3 shows that multiple materials can be added to the micro-/nanoparticle hybrid hydrogel platform. Therefore, different goals can be achieved simultaneously. Third, micro-/nanoparticles are able to compensate for hydrogel’s poor mechanical properties [38,93]. Moreover, this hybrid system is able to release diverse stowage at different stages of treatment [94]. In addition, the injectability ensures localized drug delivery, which is able to enhance the local drug concentration and avoid systemic toxicities [95]. Then, antibacterial and antioxidant properties are important for safety in vivo [96,97]. Accordingly, the micro-/nanoparticle hybrid hydrogel platform is suitable for biomedicine and tissue engineering. Currently, it is popular with research on drug delivery [39,85,98], tumors [99,100,101,102], tissue repair and regeneration [103], and wound healing [94]. Obviously, the hybrid system is promising and effective in the treatment of diseases. 

## 3. Micro-/Nanoparticle Hybrid Hydrogel Platform Applied in Articular Cartilage-Related Diseases

As for articular cartilage-related diseases, the cartilage extracellular matrix (ECM) plays a strong part in their occurrence and development. The abnormity and damage of the ECM are closely correlated with cartilage injury and interrelated diseases [104]. Aging is a factor in a break in homeostasis in the cartilage ECM [105]. Consequently, the ECM can be a signal to imply the risk of the occurrence of cartilage-related diseases, and it is necessary to prevent these and provide corresponding treatment at every stage of the disease. According to the function of the ECM, some attempts have been made to prevent OA [106,107]. 

Articular cartilage-related diseases are often accompanied by varying degrees of cartilage damage and some symptoms. Therefore, the main manifestation of diseases cannot be neglected. Cartilage degeneration [108,109] and defect [110] need to be more carefully considered as two main factors in articular cartilage-related diseases. They are also related to the occurrence of severe pain, inflammation, and dysfunction, which may seriously impact daily life. Due to the outstanding properties of the micro-/nanoparticle hybrid hydrogel platform, articular cartilage-related diseases are promising for treatment and recovery. Figure 4 shows that it is mainly involved in the symptomatic treatment of pain, inflammation, swelling, and dysfunction, as well as articular cartilage regeneration for cartilage defect. Table 1 exhibits a summary of some relevant research.

### 3.1. Micro-/Nanoparticle Hybrid Hydrogel Platform in Symptomatic Treatment 

Articular cartilage is susceptible to diseases on account of abrasion and coloboma mainly led by constant movement of the joints. Furthermore, trauma, tumor, and other reasons can also cause cartilage diseases. Additionally, the damage of cartilage causes the occurrence of articular cartilage-related diseases. Meanwhile, it can be a symptom of diseases such as OA and RA [118,119]. Additionally, patients suffer from severe pain, inflammation, and dysfunction. As a consequence, timely treatments are needed. 

**Figure 5 gels-07-00155-f005:**
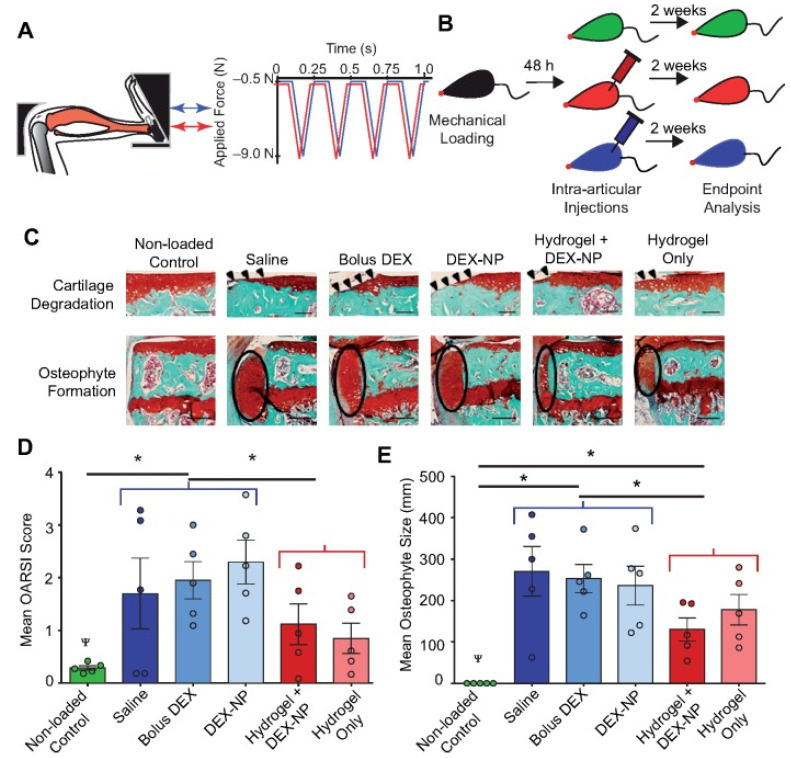
Injectable mechanical pillow-attenuated in vivo cartilage damage and osteophyte formation following the development of load-induced osteoarthritis. (**A**) Mouse tibial cyclic compression model. Schematic of tibia positioned in loading device, ready for in vivo axial loading to be applied. (**B**) Schematic of the duration of loading, intra-articular injections, and end-point analysis. The five injection formulations were saline, bolus DEX (5 mg/mL in PBS), DEX-loaded PLGA nanoparticles (8 mg/mL), hydrogel with DEX-loaded PLGA nanoparticles, and hydrogel alone. Hydrogel groups were 10% *w*/*v* PEG-4MAL with a 1:1 DTT:VPM ratio. The hydrogel group with DEX nanoparticles was ~10% nanoparticles. (**C**) Safranin O-fast green-stained images of the non-loaded vs. loaded limbs (peak load 9.0 N), indicating cartilage erosion (arrowheads) and osteophyte (ellipses) after 2 weeks of loading in the posterior aspect of the medial tibial plateau. (**D**) The mean OARSI scores of cartilage in the medial tibial plateau and (**E**) mean medial–lateral width of the osteophyte from three representative sections in the joint (posterior, middle, and anterior). n = 5 mice/group. Red groups = non-hydrogel injections; blue groups = hydrogel-containing injections. Ψ *p* < 0.05 for loading; * *p* < 0.05 for hydrogel vs. no hydrogel nested by loading. Cartilage scale bars = 100 μm. Osteophyte scale bars = 200 μm. This figure was published in [120]—copyright, Elsevier (2019). Permission to share the material has been granted.

In consideration of the movement of joints, the mechanical properties should be ensured. A composite hydrogel system made up of silated hydroxypropylmethyl cellulose (Si-HPMC) and mixed with laponites (XLG) was designed to apply to cartilage tissue engineering. This system allowed an injection into the injured region in vivo to steadily sustain the formation of cartilage tissue [121]. In addition, a nanoparticle hybrid hydrogel platform composed of core-shell nanoparticles, and hydrophobic association hydrogels (MHA gels), presented excellent mechanical properties for articular cartilage repair [122]. Bovine serum albumin (BSA) was embedded in chitosan microspheres (CMs) and released slowly through microspheres combined with injectable CMC-OCS hydrogels. It was also able to transport articular chondrocytes to help cartilage repair [123]. Researchers also designed a suitable micro-/nanoparticle hybrid hydrogel platform to relieve troublesome symptoms. Holyoak et al. [120] created a four-arm maleimide-functionalized polyethylene glycol (PEG-4MAL) hydrogel system loaded with fluorescent fluorescein isothiocyanate (FITC) polystyrene nanoparticles containing dexamethasone as a mechanical pillow (Figure 5). It showed great mechanical properties to relieve stress and protect knee-joint cartilage. This system was also able to release dexamethasone according to the needs of the patient and played a role in anti-inflammatory action at the appropriate times.

Due to immune dysregulation, chronic inflammation and irreversible joint and organ damage may be caused [124]; classical symptoms of immune dysregulation and chronic inflammation are expected to be resolved in the near future in order to avoid further aggravation of the disease. Methotrexate (MTX) is known as a first-line drug for the clinical treatment of RA due to its desirable anti-inflammatory action and ability to enhance the effectiveness of biological agents [125]. A sort of nanoparticle-loaded indomethacin (IND), MTX, and a small-interfering RNA (targeting MMP-9) were added to the hydrogel to form a suitable nanoparticle hybrid hydrogel platform for RA treatment. Relieving swelling, reducing inflammation, and reversing the damage of articular cartilage can be achieved at the same time by injecting the hybrid hydrogel platform into the diseased joint [126]. Yin et al. [127] designed a nanoparticle hybrid hydrogel platform for the synergistic treatment of RA. In this study, the hybrid system was made of temperature-sensitive hydrogel (D-NGel) and nanoparticles (D-NPs), which were loaded with indomethacin (IND) and MTX—specifically, the long-acting hydrogel formed after intra-articular injection. In vivo, it was helpful for reducing joint swelling, inflammation, and bone erosion.

As for articular cartilage-related diseases, including OA and RA, some symptoms, such as inflammation, pain, and swelling, are similar. Therefore, some researchers focused on specific symptoms instead of the type of disease. Platelet lysate (PL), a mixture of growth factors, was designed to load in nanoparticles combined with thermosensitive poly(D,L-lactide)-poly(ethylene glycol)-poly(D,L-lactide) (PLEL) hydrogel. The well-designed system was able to decrease the inflammatory response, avoid the degeneration of cartilage in the early stages of diseases, and enhance the cartilage repair (Figure 6) [128]. In order to make useful, long-acting drugs to protect cartilage and its surrounding tissue, as well as control inflammation, agarose hydrogel and poly(lactic-co-glycolic) acid (PLGA) microspheres were loaded with dexamethasone (DEX) and bonded together to aid the continuous release of the effective drug [129]. The severe pain that patients suffered from should also be alleviated. Long-acting analgesic could be achieved by a microsphere hybrid hydrogel platform. On account of the loading of bupivacaine (BUP) and DEX, the analgesia time was obviously prolonged [86]. Furthermore, the micro-/nanosphere hybrid hydrogel platform can also be used as an applicable implant for orthopedic surgery. Chondrocytes were found to survive and grow in the microsphere hybrid hydrogel platform that comprised cartilaginous hydrogel and PLGA microspheres [130]. Asgari et al. attempted to load curcumin (Cur) in hydrogel to deactivate inflammation, and PLGA microparticles covering kartogenin were embedded within it. Through in vivo experiments, this injectable hybrid hydrogel platform has also been shown to help cartilage regeneration [131].

### 3.2. Micro-/Nanoparticle Hybrid Hydrogel Platform in Cartilage Defect 

It is difficult to heal cartilage defect and regain cartilage function. To solve this problem, researchers focused on two methods, cartilage repair and cartilage regeneration. Some attempted to use other appropriate materials to replace the missing cartilage [132]. Others have been dedicated to regenerating cartilage where it is missing using suitable materials. Although many problems need to be taken into account, biocompatibility, cell attachment, loading capacity, mechanical ability, and other properties are needed. Fortunately, the micro-/nanoparticle hybrid hydrogel platform can satisfy these requirements. It is able to carry the required materials and act as a scaffold for cartilage repair and regeneration. 

Regarding cartilage repair, material selection of the micro-/nanoparticle hybrid hydrogel platform is important. Meanwhile, structural design is equally important. The micro-/nanoparticle hybrid hydrogel platform is suggested to be a more suitable scaffold for cartilage tissue engineering. Researchers prepared natural polymers including maleilated chitosan (MCS) and methacrylated silk fibroin (MSF) micro-/nanoparticles and added them into hydrogel. Outstanding biocompatibility and cell adhesion proved that it had great potential for cartilage repair [133]. Liao et al. [134] designed an injectable MPs/Alg-chondrocyte system that consisted of porous poly(ε-caprolactone)−β-poly(ethylene glycol)−β-poly(ε-caprolactone) microspheres (MPs/Alg) and three-dimensional (3D) alginate hydrogel. Many experiments have suggested its usefulness in repairing cartilage defects.

Compared to cartilage repair, cartilage regeneration seems more difficult to achieve. Currently, a great number of studies pay attention to promoting cartilage regeneration by different thoughts. First, as a scaffold, the ability to promote cell growth and differentiation is important. Hydroxyapatite nanoparticle-crosslinked peptide hydrogels designed by Xu et al. [135] presented great potential in cartilage regeneration. Then, choosing a suitable material that possesses the ability to regenerate cartilage to add to a micro-/nanoparticle hybrid hydrogel platform is important. Through a large amount of research, activated skeletal stem cells and mesenchymal stem cells (MSCs) were confirmed to be appropriate to add to the micro-/nanoparticle hybrid hydrogel platform [136]. 

Kouhi et al. [137] developed an injectable gellan gum/lignocellulose nanofibril hydrogel system mixed with melatonin-loaded forsterite nanoparticles. It showed high mechanical properties and the capability of sustainable drug release. Additionally, this system was able to promote cell adhesion and facilitate proliferation and gene expression, which suggest its appropriateness for cartilage regeneration. Lua et al. [138] attempted to combine collagen hydrogel with carbon dot nanoparticles (CD NPs), which successfully enhance the difficulty to adapt to chondrogenesis (Figure 7). The injectable hybrid hydrogel platform was able to produce reactive oxygen species (ROS) to perfect chondrogenic differentiation and achieve the regeneration of cartilage. Additionally, eggshell microparticles (ESP) were dispersed in hydrogel to obtain better mechanical properties [139].

Drugs, growth factors, and stem cells, as well as chondrocytes, are available to add to the micro-/nanoparticle hybrid hydrogel platform to promote the regeneration of cartilage. Therefore, many researchers are committed to finding means for applicable stowage to load in the system for cartilage regeneration.

In order to regenerate cartilage, it has also been attempted to load drugs that are favorable for cartilage regeneration in the micro-/nanoparticle hybrid hydrogel platform to achieve continuous release. Naghizadeh et al. [140] designed an injectable microparticle hybrid hydrogel platform with the addition of melatonin and methylprednisolone. This system provided a good environment for cell growth, which means that cartilage regeneration is expected to be realized. 

Fan et al. [141] designed a nanoparticle hybrid hydrogel platform that loaded kartogenin (KGN) and transforming growth factor β3 (TGF-β3) to stimulate the formation of cartilage. This system mainly acted on the endogenous mesenchymal stem cells (MSCs). TGF-β3-loaded PLGA nanoparticles were added to alginate-poly(acrylamide) hydrogel to create a hybrid system. Then, the controlled release of TGF-β3 showed the effect of enhancing cartilage regeneration (Figure 8) [142]. Furthermore, chondroitin sulfate (ChS) was used to relieve pain and promote cartilage regeneration in joint disease. Radhakrishnan et al. [143] add Ghs-loaded microparticles to porous injectable biphasic semi-interpenetrating polymer network (SIPN) hydrogel for articular hyaline cartilage regeneration.

The micro-/nanoparticle hybrid hydrogel platform is supposed to provide a proper metabolic environment for cells to grow. Therefore, a microparticle hybrid hydrogel platform that was able to release oxygen was created. Researchers placed calcium peroxide in PLA microparticles and dispersed them in the hydrogel. The oxygen released from the system can satisfy the requirements of cell metabolism at the cartilage-to-bone interface [114].

Additionally, magnetic nanocomposite hydrogels were suggested to be applied in cartilage tissue engineering [144]. Huang et al. [145] attempted to add Fe_3_O_4_ magnetic nanoparticles to hydrogel to support the adhesion, growth, and proliferation of MSCs due to the effects of the pulsed electromagnetic field. It was able to promote the chondrogenic differentiation of BMSCs and showed a promising opportunity for the clinical treatment of articular cartilage defects. In addition, magnetic poly(lactic-co-glycolic acid) microsphere-gelatin hydrogel designed by Wang et al. [146] was supposed to be able to promote bone repair.

Furthermore, the micro-/nanoparticle hybrid hydrogel platform was expected to be equipped with multiple effects to realize more than one purpose at once. Atoufi et al. [111] created an injectable thermosensitive micro-/nanoparticle hybrid hydrogel platform with triple effects. In this platform, PLGA-ACH micro-/nanoparticles contained melatonin, and PNIPAM/hyaluronic acid hydrogel was thermosensitive. Improving mechanical properties, controlling releasing chondrogenic small molecule melatonin, and reducing the dehydration and shrinkage of hydrogels could be realized through the well-designed hybrid system. Therefore, it had great potential for applications in cartilage tissue engineering. 

In addition, there are some current studies focused on improving the performance of micro-/nanoparticle hybrid hydrogels to make it easier to use them in cartilage regeneration. In order to improve the properties of the drug-loaded nanoparticle hybrid hydrogel platform, Leea et al. [147] devised a method to promote stem-cell adherence to the system by means of cold atmospheric plasma (CAP) treatment.

## 4. Prospect and Challenges

The micro-/nanoparticle hybrid hydrogel platform shows tremendous potential in the treatment of articular cartilage-related diseases. Therefore, it is a promising application for cartilage tissue engineering and solving intractable clinical problems. It is certain that the micro-/nanoparticle hybrid hydrogel platform presents great superiority in many ways. Nevertheless, there are still some issues to be considered. First, this hybrid system should be designed to match the complex microenvironment in vivo, including friction, pH and temperature, and its safety should be guaranteed. Nanoparticles may pose toxicity issues through distribution and deposition in cells and different organ systems, such as the lungs, liver, and brain. The route of administration and exposure can influence the toxicity of NPs, and the toxicity of NPs is mainly determined by size, surface area, surface charge, and aggregation state [148]. Therefore, the toxicity of NPs should be considered and analyzed before use. Accordingly, experiments in vitro and in vivo that aim to improve safety and eliminate influencing factors are warranted.

Second, controlled release is an outstanding property of the system. However, the release mechanism and effectiveness remain unclear. Owing to the requirement of a specific quantity of stowage for the system, especially in drugs, whose dosage is closely related to security and efficacy, the release mechanism and effectiveness require rigorous studies. On the other hand, ensuring the difference between micro-/nanoparticles and hydrogel in the micro-/nanoparticle hybrid hydrogel platform is needed. Only in this way can we solve the issue of loading different materials into befitting places and controlling the sequence of releases. Additionally, due to the diversity of micro-/nanoparticles and hydrogels, it becomes difficult to choose the right combination of materials in order to maximize the performance of the hybrid system.

In addition, the kinds of materials that are most appropriate to be loaded in the system should be studied in detail. It is not only necessary to conduct thorough experiments, but the study of mechanisms also cannot be ignored. Suitable cells with desirable effects are expected to be added. Some studies suggest that articular cartilage morphogenesis, cells that are involved in articular cartilage development, may provide insights into cell types and sources of joint injury [149]. Focusing on mechanisms of articular cartilage development and damage may play a significant role in the treatment of articular cartilage-related diseases and cartilage regeneration and provide useful developments.

Furthermore, articular cartilage diseases may cause damage to adjacent bone tissue. Problems of cartilage defects and bone defects are also expected to be solved by the micro-/nanoparticle hybrid hydrogel platform. Therefore, some relevant studies that focused on bone defect repair can be highlighted. Bowen tan’s team designed an injectable curcumin-microsphere/IR820 coloaded hybrid methylcellulose hydrogel (Cur-MP/IR820 gel) platform, which solved the problem of bone defects caused by osteosarcoma by promoting bone reconstruction [100]. Additionally, as a soft tissue, articular capsules can also be damaged. Therefore, it is equally essential to solve this problem. SIS powder (SISP)/chitosan chloride (CSCl)-β-glycerol phosphate (GP)-hydroxyethyl cellulose (HEC) hydrogel was produced and has potential in soft tissue regeneration [150].

Furthermore, previous experiments mainly pay close attention to knee diseases. However, other joint diseases also warrant consideration. The study of the temporomandibular joint lacks suitable animal models [151]. As a special joint, the previously mentioned temporomandibular joint is easily damaged in the condyle cartilage, with no discomfort until the condition becomes highly serious. Compared to knee disease, the temporomandibular joint is harder to treat due to the particularity of its position and structure. Due to the superiority of the micro-/nanoparticle hybrid hydrogel platform in other articular cartilage-related diseases, it may be appropriate to apply in temporomandibular joint diseases, such as temporomandibular joint disorders (TMD) [152], temporomandibular joint osteoarthritis, (TMJOA) [153], and temporomandibular joint ankylosis. There are some studies on cartilage tissue engineering [154,155] that can provide an applicable basis in the micro-/nanoparticle hybrid hydrogel platform for the treatment of the temporomandibular joint. Through experiments, exosomes (Exos) of stem cells from human exfoliated deciduous teeth (SHEDs) were claimed to protect TMJ chondrocytes from inflammation that is mainly caused by TMJOA [156]. 

In summary, in order to improve the properties of the micro-/nanoparticle hybrid hydrogel platform for use in the treatment of articular cartilage-related diseases, the prospect that should be realized is described as follows: (a) improving safety—it might be helpful to wrap it in secure materials, modify the surface properties of materials that may harm human health, or set up a strain-like structure to avoid harmful substances entering blood circulation; (b) increasing adaptability—improving the lubricity of the surface of materials according to the features of joints and performing vitro experiments under the same conditions as in vivo; (c) studying the mechanism clearly; (d) choosing accurate materials by establishing some conditions such as biocompatibility and mechanical properties and comparing them according to those conditions; (e) solving the problem of adjacent bone destruction; and (f) addressing broader joint diseases.

Additionally, there are still some unresolved challenges. A large amount of effort is required based on the current research status. Before being put into use, the micro-/nanoparticle hybrid hydrogel platform still requires sufficient clinical tests for the hybrid system to be proven to be completely safe for the body and highly effective for these diseases. Moreover, an accurate operating guide should be produced through repeated experiments to instruct the doctor on dosage, duration, and interval. Additionally, we cannot ignore the appropriate methods of therapeutic effect evaluation for humans. Unlike animal experiments, human trials require noninvasive and entirely safe methods to check the efficacy of treatment. At the same time, technical problems of the micro-/nanoparticle hybrid hydrogel platform also exist. We ignored the fact that the internal structure and effect of the micro-/nanoparticle hybrid hydrogel platform may change under the influence of volume and friction. Therefore, guaranteeing that micro-/nanoparticles are uniformly embedded within the hydrogel after injection, extrusion, or friction caused by activity and that the dose of drug release is stable should be resolved. If these issues are resolved completely, the micro-/nanoparticle hybrid hydrogel platform is promising for wide use in the clinic.

## 5. Conclusions

In this review, we summarize the latest progress of the micro-/nanoparticle hybrid hydrogel platform in the treatment of different types of joint diseases that can cause articular cartilage injury. Due to great biocompatibility, mechanical strength, and syringeability, micro-/nanoparticle hybrid hydrogel presents superiority and a desirable effect on the treatment of articular cartilage-related diseases. This platform shows great potential in symptom control and cartilage repair and regeneration. This review provides a useful strategy for the treatment of articular cartilage-related diseases in cartilage tissue engineering, which may provide researchers with new ideas for clinical treatment.

## Figures and Tables

**Figure 1 gels-07-00155-f001:**
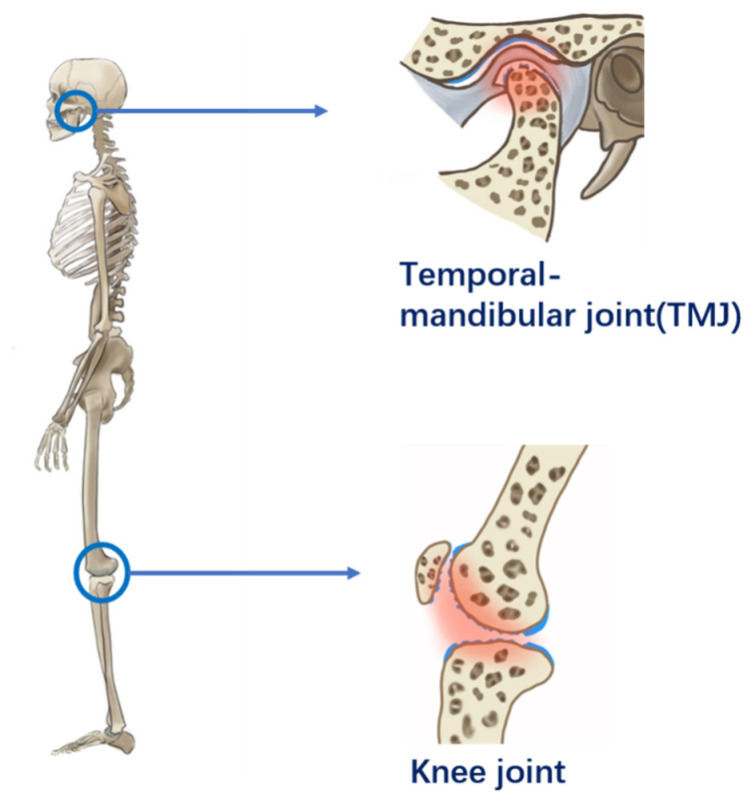
Articular cartilage defects in the temporomandibular joint (TMJ) and knee joint.

**Figure 2 gels-07-00155-f002:**
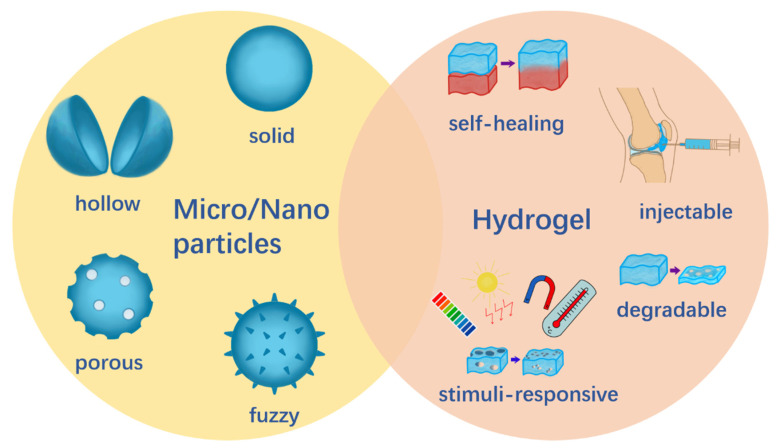
The variety of micro-/nanoparticles and hydrogels.

**Figure 3 gels-07-00155-f003:**
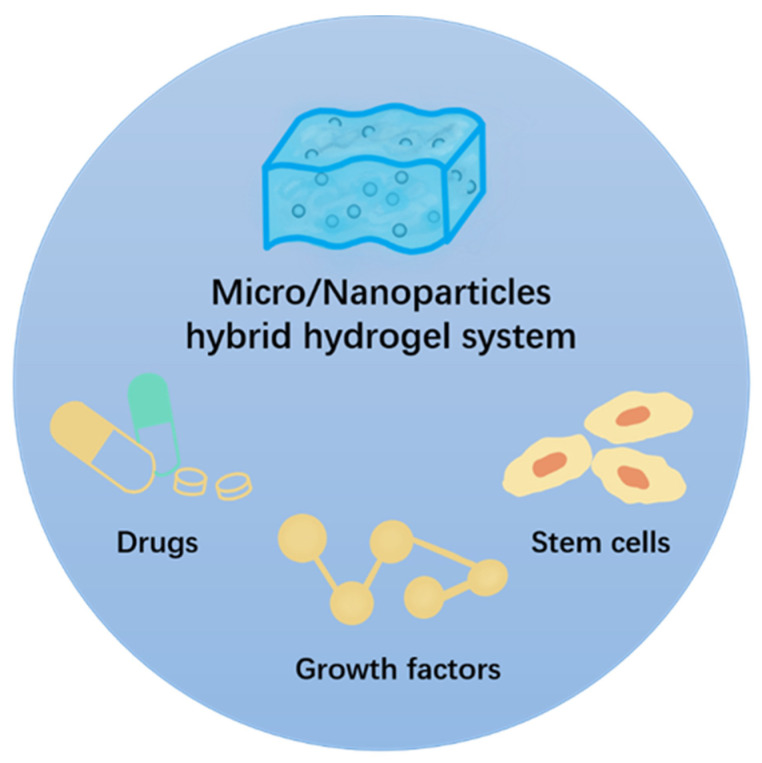
Drugs, growth factors, and stem cells can be added to the micro-/nanoparticle hybrid hydrogel platform.

**Figure 4 gels-07-00155-f004:**
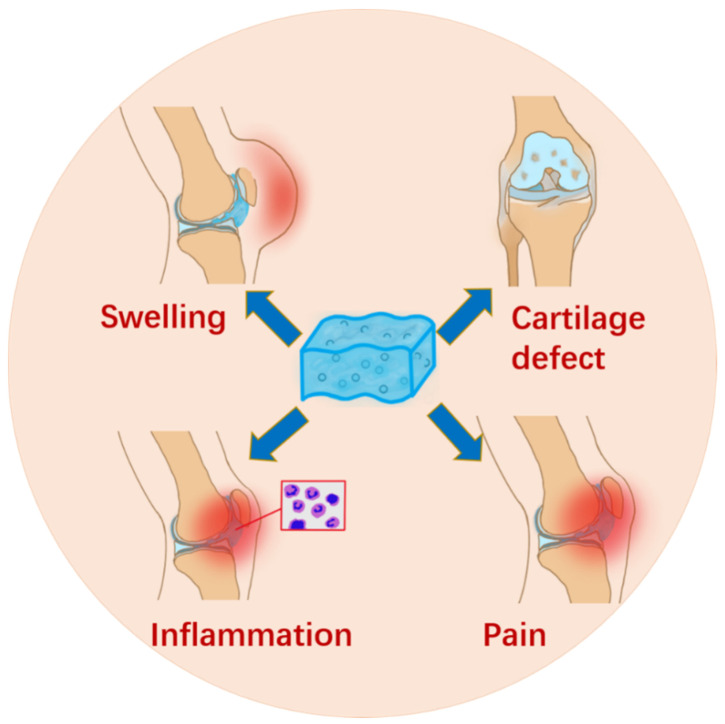
Micro-/nanoparticle hybrid hydrogel platform applied in articular cartilage-related diseases.

**Figure 6 gels-07-00155-f006:**
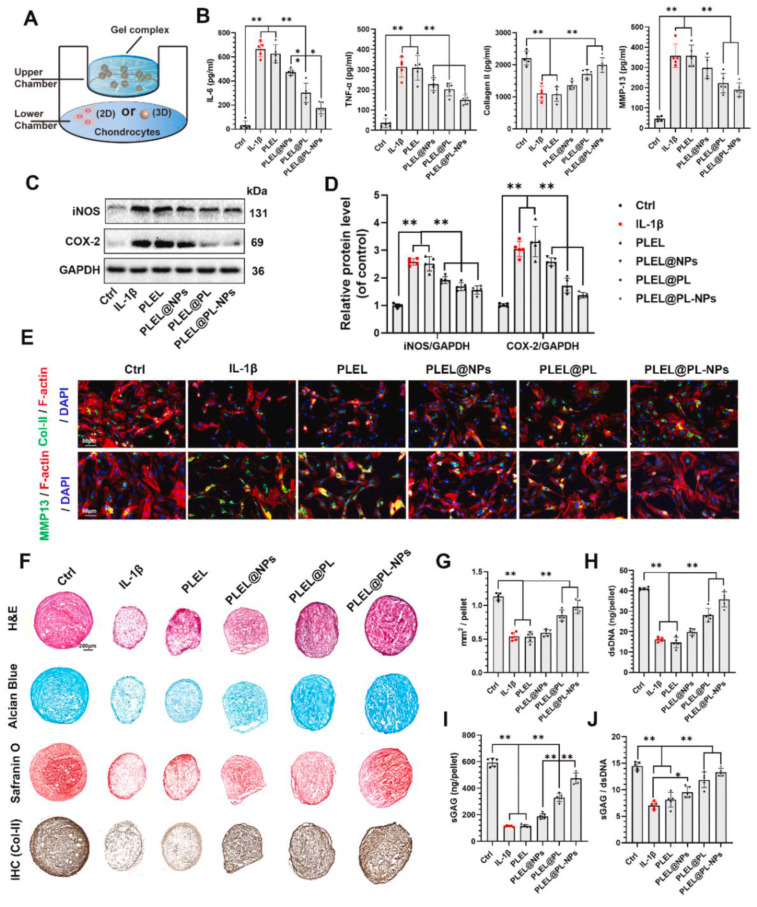
In vitro anti-inflammatory performance of the PLEL@PL-NP system. (**A**) Diagram of the co-cultural method for gel (upper) and chondrocytes (lower, 2D or 3D culture). (**B**) Secretory protein levels of IL-6, TNF-α, collagen II, and MMP13 in chondrocyte culture medium with different treatment after 14 days of co-culture. (**C**,**D**) Protein levels of iNOS and COX-2 in chondrocytes after 14 days of co-culture with different treatments. (**E**) Immunofluorescence staining of cells co-cultured with different treatments as indicated at day 14. (**F**) Histological staining (H&E), GAG staining (Alcian blue and Safranin O), and immunohistochemistry staining of collagen II in pellets with different treatments, as indicated at day 21. (**G**) Quantification of pellet size. (**H**) Quantification of dsDNA content. (**I**) Quantification of sGAG content. (**J**) Quantification of the ratio of sGAG/dsDNA. All quantitative results are presented as means ± SD of five replicate experiments. * *p* < 0.05, ** *p* < 0.01. (For interpretation of the references to color in this figure legend, the reader is referred to the web version of this article.) This figure was published in [128]—copyright, Elsevier (2021). Permission to share the material has been granted.

**Figure 7 gels-07-00155-f007:**
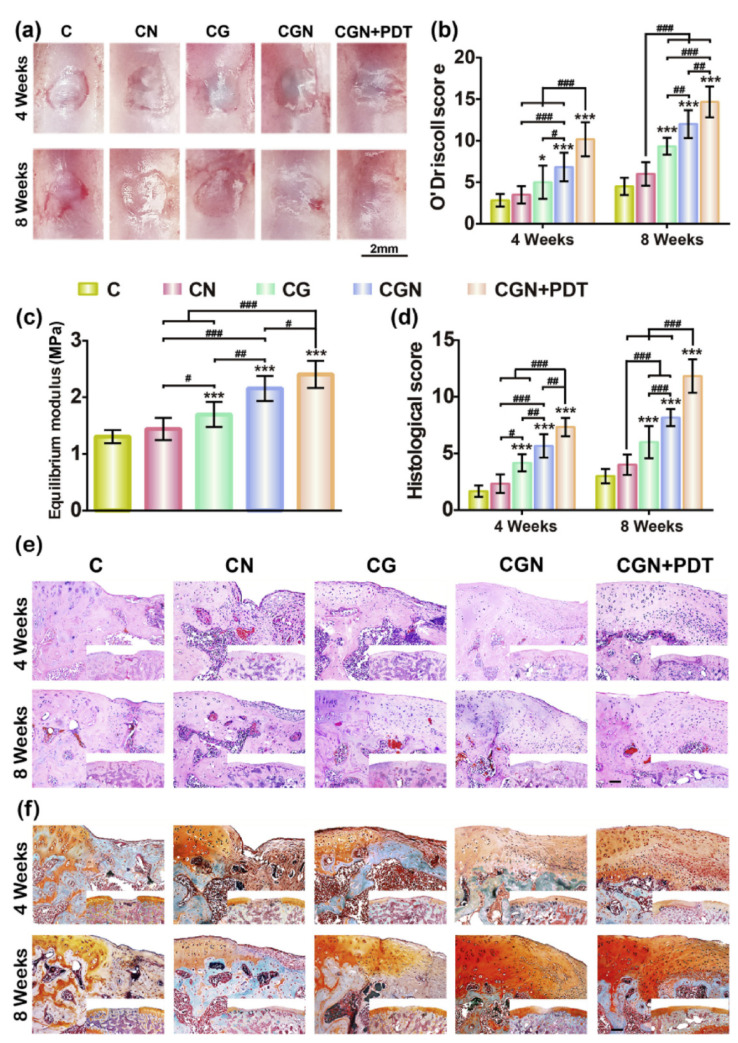
Cartilage regeneration promoted by PDT after implantation with CGN. (**a**,**b**) Gross macroscopic observation (**a**) and O’Driscoll score (**b**) of engineered cartilage (scale bar = 2 mm). (**c**) Mechanical evaluation of the engineered cartilage after 8 weeks of treatment. (**d**) Histological score of the engineered cartilage after 4 and 8 weeks of treatment. (**e**,**f**) HE (**e**) and Safranin O/fast green (**f**) staining of the engineered cartilage (scale bar = 200 μm). Mean ± SD, n = 6; *, # indicate *p* < 0.05, ## indicates *p* < 0.01, ***, ### indicates *p* < 0.001. (C = collagen; CN = collagen mixed with CD NPs; CG = collagen crosslinked with genipin; CGN = collagen crosslinked with genipin and CD NPs; CGN + PDT=CGN after 808 nm laser irradiation at a power density of 8.3 mW/cm^2^ for 3 min.) (For interpretation of the references to color in this figure legend, the reader is referred to the web version of this article.) This figure was published in [138]—copyright, Elsevier (2019). Permission to share the material has been granted.

**Figure 8 gels-07-00155-f008:**
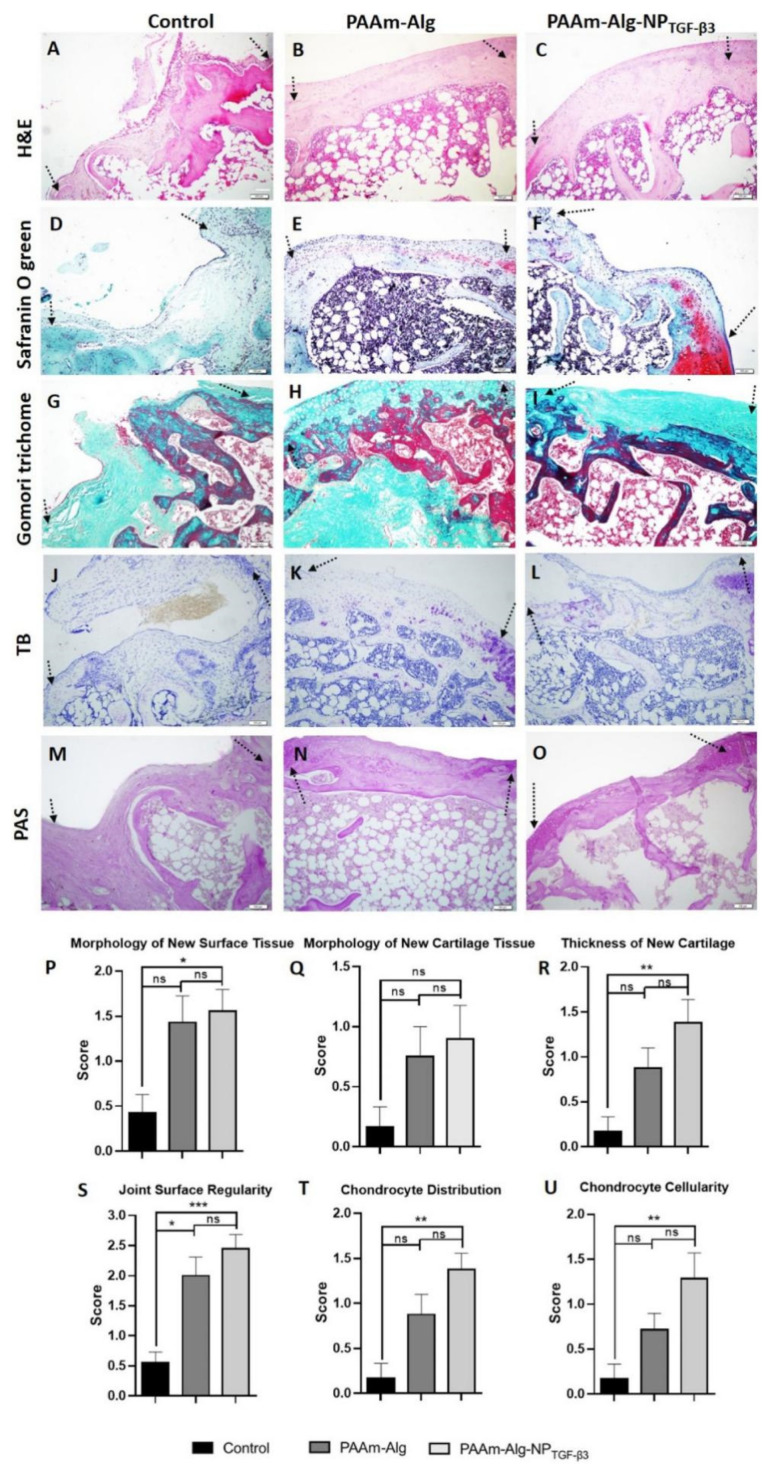
In vivo representative photomicrographs of tissue sections in the experimental groups stained with (**A**–**C**) H&E; (**D**–**F**) Safranin O/fast green, (**G**–**I**) Gomori’s trichrome stains, TB (**J**–**L**), and PAS (**M**–**O**). (**A**,**D**,**G**) Control group demonstrates mostly fibrous tissue with limited tissue repair in the chondral region. (**B**,**E**,**H**) PAAm-Alg group shows fibrous tissue with few fibroblast and chondrocyte-like cells in the joint surface chondral regions. (**E**) Fibrous tissue in the deeper regions is still observed in some rats. (**C**,**F**,**I**) PAAm-Alg-NPTGF-β3 group reveals fibrous tissue with some fibrocartilage and hyaline cartilage groups in the chondral regions. PAAm-Alg and PAAm-Alg-NPTGF-β3 groups demonstrated higher (**K**,**L**) glycosaminoglycan deposition and (**N**,**O**) glycoprotein and proteoglycan contents compared to control group (the arrows indicate the margins of the defect area; scale bars: 100 μm). The histological lesion scores in regard to the morphology of (**P**) new surfaces, (**Q**) cartilage tissues, (**R**) tissue thickness, (**S**) surface regularity, (**T**) chondrocyte distribution, and (**U**) cellularity. ns > 0.05, * *p* < 0.05, ** *p* < 0.01, *** *p* < 0.001; Kruskal–Wallis test. This figure was published in [142]—copyright, Elsevier (2021). Permission to share the material has been granted.

**Table 1 gels-07-00155-t001:** Application of micro-/nanoparticle hybrid hydrogel platform in articular cartilage-related diseases.

Micro-/Nanoparticles	Hydrogel	Property	Stowage	Application	Effect	Ref.
Chitosan-acrylic acid-coated PLGA (ACH-PLGA) micro-/nanoparticles	PNIPAM/hyaluronic acid	injectable, thermosensitive, biocompatibility	melatonin	cartilage tissue engineering	sustained drug release	[111]
Hydroxyapatite (HAp) microparticles	RGD-alginate	injectable, viscoelastic	Sr	bone repair	osteogenic, anti-osteoclastogenic and immunomodulatory	[112]
Chitosan (CS) nanoparticles	silk fibroin (SF)	biocompatibility	TGFβ1@CS, BMP-2@SF	repair knee joint cartilage defects	release TGF-β1 and BMP-2, promoted chondrogenic ability of BMSCs	[113]
Polylactic acid (PLA) microparticles	hydrogel of functionalized pectin and fibroin	\	calcium peroxide	a promoting step for regeneration of cartilage-to-bone interface	oxygenation of the cartilage-to-bone interface	[114]
Hyaluronic acid (HA)/chitosan-poly(dioxanone)(CH-PDO) complex nanoparticles	ALG-POL/SF dual network	elastic, tough and strong	bone morphogenic protein-7 (BMP-7)	cartilage tissue engineering	controlled release of BMP-7	[115]
CNT nanoparticles	polyacrylamide (PAM)	bioactivity and cytocompatibility	TiO2	cartilage repair	cartilage replacement	[116]
Poly(lactic acid) (PLA)/methoxy-polyethylene glycol-poly(δdecalactone) (mPEG−PDL) microparticles	poly(PEGMA) Copolymer	thermoresponsive, bioadhesive	triamcinolone acetonide (TA)	RA	anti-inflammatory	[117]

## Data Availability

Not Applicable.

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
