# Peer review of "Effect of Micro-/Nanoparticle Hybrid Hydrogel Platform on the Treatment of Articular Cartilage-Related Diseases"

_gels, 2021, doi:10.3390/gels7040155_

Round 1
Reviewer 1 Report
The authors present a review of recent literature in the area of application of nano/ micro particle/ hydrogel systems for the treatment of articular cartilage diseases. The subject falls within the scope of the journal, but the following should be attended to before the manuscript is considered for publication:
- Line 32: Replace "Whats worse" with Moreover.
- Line 50-52: Strong claims are made in these lines. Please support these claims with appropriate references.
- Lines 91-92: What's the relationship between nano/ micro particles and hydrogels? Are the authors implying that the particles are embedded within the hydrogels? Please clarify this in the text.
- Lines 119-133: Hydrogels are known carriers of drugs and have been extensively studied for drug release applications. Controlled or sustained release remains a major short coming for hydrogels. Drug loaded nano/ micro particles embedded in hydrogels have been used to overcome this challenge. This, with any other advantages offered by the nano/ micro particles, should be clearly discussed in the text. Also, please provide appropriate references.
- Section 3.1: Mechanical properties of hydrogels have been very briefly discussed here. This discussion should be extended to include the effect of nano/ micro particles on the tribology of hydrogels considered for articular cartilage repair.
- Do nano particles pose toxicity issues? Please discuss this subject as well.
Reviewer 2 Report
Please simply and clearly address the prospects for improving the property of the micro/nanoparticles hybrid hydrogel platform to offer meaningful new ideas for effective and accurate treatment for articular cartilage-related diseases.
The meaningful discussion of the platform technology would be the most important part of a good review paper. Please focus on the technical problems of the micro/nanoparticles hybrid hydrogel platform to discuss what has been missed, what is the key issue to be solved next.
Round 2
Reviewer 1 Report
The authors have attended satisfactorily to the corrections and improvements suggested by the reviewer. The manuscript has been improved significantly and can now be considered for publication. Minor English language corrections will still help improve the quality of the manuscript.
Reviewer 2 Report
Can be published now.